

# Textural quality, growth parameters and oxidative responses in Nile tilapia (*Oreochromis niloticus*) fed faba bean water extract diet

Yichao Li[1,2,*], Junming Zhang[1,*], Bing Fu[1], Jun Xie[1], Guangjun Wang[1], Jingjing Tian[1], Yun Xia[1] and Ermeng Yu[1]

[1] Pearl River Fisheries Research Institute of CAFS, Guangzhou, China
[2] Key Laboratory of Aquatic Animal Immune Technology of Guangdong Province, Pearl River Fisheries Research Institute of CAFS, Guangzhou, China
[*] These authors contributed equally to this work.

## ABSTRACT

Texture is one of the key quality attributes used in the fresh and processed fish industry to assess product quality and consumer acceptability. To improve the textural quality of tilapia, we formulated the expanded pellet diet (EPD) and pellet diet (PD), both containing faba bean (*Vicia faba*, FB) water extract, a previously reported potential aquafeed additive to increase flesh texture. The common diet was used as a control. After Nile tilapia (*Oreochromis niloticus*) were fed three kinds of experimental diet for 120 days, muscle textural quality, growth parameters, oxidative response and immune parameters were analyzed. The results showed that there was no significant difference in the growth parameters between the three groups ($P > 0.05$). The highest measure of textural quality (hardness and chewiness) was found for the PD group, followed by the EPD and the control ($P < 0.05$). Less oxidative damage to the hepatopancreas and intestine was found in the EPD compared with the PD group, as demonstrated by the decreased levels of reactive oxygen species and increased levels of nicotinamide adenine dinucleotide and intestinal digestive enzyme activity (amylase and lipase). Taken together, this study highlights the potential usefulness in commercial settings of FB water extract for improving the textural quality of tilapia, and EPD containing faba bean water extract could be more advanced substitute for faba bean in tilapia culture in term of both effectiveness in textural quality improvement and health status enhancement compared with PD.

## INTRODUCTION

Nile tilapia (*Oreochromis niloticus*), an extensively cultured fish species, had a global production of 4.53 million tons in 2018. It provides low-cost and high-quality animal protein, especially for developing and underdeveloped regions (*FAO, 2020*). However, with the development of intensive aquaculture at high stocking densities, a reduction in flesh quality and textural properties have become one of the most important issues in the

Corresponding author
Ermeng Yu, boyem34@hotmail.com, yem@prfri.ac.cn

tilapia industry (*Wu et al., 2018*). High stocking density has been reported to exert a negative impact on growth performance and muscle quality due to chronic crowding stress in fish (*Onxayvieng et al., 2021*; *Roth, Slinde & Arildsen, 2006*). Also, the growing aquaculture production of tilapia was reported not to yield high profit to fish farmers, probably due to low price or poor muscle quality (*Dey et al., 2000*). Therefore, the production of high-quality tilapia would be promising for improving its culture efficiency and profitability.

The texture is an important quality characteristic of a fish product that is associated with consumer acceptance (*Chen et al., 2021*). Crisp grass carp (*Ctenopharyngodon idellus* C. et V) is a successful example: it is a representative carp variety with improved textural characteristics (hardness, chewiness, *etc.*) after fed solely faba bean (FB; *Vicia faba*) for 90–120 days (*Chen et al., 2020*). Its fillets have been exported to various countries in Southeast Asia and Latin America (*Yang et al., 2015*). Similar textural characteristics were also found in the tilapia fed solely FB, but FB feeding caused decreased growth rate, probably due to its poor palatability and anti-nutritional factors (*Lun et al., 2007*). Furthermore, the alcohol and ester extracts of FB were applied to tilapia culture, but they did not effectively improve the textural characteristics of tilapia (*Chen et al., 2014*), possibly because the functionally active substances were not obtained.

Previous studies have shown that the FB water extract can be used as a functional aquafeed additive to improve the textural quality of grass carp (*Ma et al., 2020a*; *Ma et al., 2020b*). However, the effect of FB water extract on the textural quality of tilapia remains unclear. Furthermore, FB water extract was only supplemented into the pellet diet (PD) of grass carp, which was formulated at moderate temperature (<55 °C) to maintain the activities of heat-sensitive active components. During the tilapia culture, the expanded pellet diet (EPD) is more commonly used compared to the PD for two reasons: (1) the EPD has higher water stability and durability; (2) EPD makes it easier for fish farmers to observe fish feeding conditions since it floats or sinks slowly (*Bolivar, Jimenez & Brown, 2006*). Thus, it deserves further investigation whether FB water extract improves the textural quality of tilapia and retains its biological efficacy in EPD throughout the intense formulating process (120 °C).

The main objective of this study was to evaluate the textural quality of tilapia fed the EPD and PD containing FB water extract. We examined the muscle textural quality (hardness, chewiness, *etc.*) and the collagen content, growth parameters, oxidative and antioxidative parameters, immune parameters and the digestive enzyme activity (lipase, amylase, and trypsin), and we performed histological analyses of the hepatopancreas and intestine. Together with the evidence from FB water extract EPD applied in a commercial setting, the present study would provides practical strategies to improve muscle textural quality in tilapia.

## MATERIALS AND METHODS

### Experimental feeds

The experimental process is showed in Fig. 1. Firstly, faba bean water extract was obtained using methods from a previous study (*Ma et al., 2020a*). In short, FB was shelled and

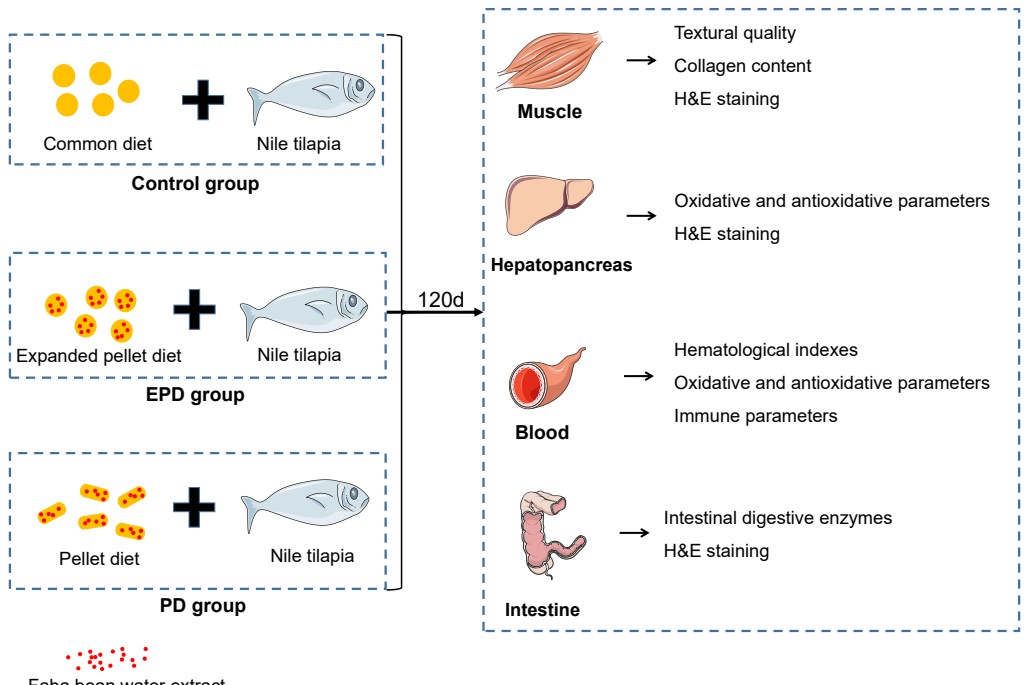

**Figure 1** **The experimental process.**

grounded into 60-$\mu$m powder and mixed with distilled water. The pH was adjusted pH to 9.0 with saturated calcium hydroxide solution. After ultrasonic processing in a sonication bath at 40 kHz for 1 h at 45 °C, the mixtures were stood for 30 min and were filtered with gauze to obtain the supernatant. Subsequently, the FB water extract was obtained by concentrating the supernatant at 5,030 g for 10 min. According to preliminary experiments and aquaculture experiences, the FB water extract was added at the ratio of 80 g/1,000 g to the experimental diets.

The PD with FB water extract was prepared as follows. First, the ingredients powder was mixed with approximately 30% distilled water (by volume), and then the soft pellets were extruded into a thin strip of 6-mm diameter by an electric smashing machine TH-18 (Tianqi Shengshi, Yongkang, China). Subsequently, the pellets were cut into lengths of 3-mm, air-dried at 50 °C for 6 h, and stored at 4 °C. To maintain the activity of heat-sensitive components, the extracting and formulating temperature of the PD was kept below 55 °C. The EPD containing FB water extract was prepared as follows. Before manufacture, all ingredients were passed through a hammer mill to achieve a particle size of <1.0 mm, and they were then mixed and blended. Subsequently, EPD was produced in an expander (TSE65S; Yang Gong S&T, China) at high temperature (120 °C). Lastly, the expanded pellets were dried at 50 °C for 6 h. The expanded pellets had a nominal diameter of five mm and a length of three mm. The common diet (a kind of EPD with no FB water extract) was used as the control. The feed ingredients and nutritional components of three

**Table 1**  Ingredients and nutritive components of experimental diets and FBW.

| | Experimental diets | | | FBW |
| --- | --- | --- | --- | --- |
| | **Control** | **EPD** | **PD** | |
| FBW (g/kg) | – | 80 | 80 | – |
| Fish meal (g/kg) | 60 | 60 | 60 | – |
| Chicken meal (g/kg) | 30 | 30 | 30 | – |
| Soybean meal (g/kg) | 290 | 290 | 290 | – |
| Rapeseed meal (g/kg) | 240 | 200 | 200 | – |
| Rice bran (g/kg) | 126 | 86 | 86 | – |
| Low-gluten flour (g/kg) | 190 | 190 | 190 | – |
| Soybean oil (g/kg) | 30 | 35 | 35 | – |
| Bentonite (g/kg) | 5 | 0 | 0 | – |
| Omnivorous fish premix (g/kg) | 10 | 10 | 10 | – |
| Calcium dihydrogen phosphate (g/kg) | 15 | 15 | 15 | – |
| Choline chloride (g/kg) | 2 | 2 | 2 | – |
| Common salt (g/kg) | 2 | 2 | 2 | – |
| Crude Protein (g/100g) | 30.48 | 30.48 | 30.52 | 38.1 |
| Crude Fat (g/100g) | 6.14 | 6.14 | 6.17 | 34.9 |
| Moisture (g/100g) | 10.80 | 10.79 | 10.86 | 9.3 |

Notes.

Control, common diet group; EPD, expanded pellet diet group; PD, pellet diet group; FBW, faba bean water extract.

experimental diets and FB water extract are shown in Table 1. The composition of FB water extract is already known (crude protein 38.1%, crude fat 34.9%, moisture 9.3%).

## Fish culture

Healthy Nile tilapia were purchased from an aquaculture farm in Zhuhai, Guangdong Province, China. The fish were first temporarily cultured in a cement pond (5 m × 5 m × 1 m) for one week and the feed amount each day was $2.5 \pm 0.5\%$ of fish weight. Subsequently, to avoid larvae reproduction triggered by a mixed culture of male and female tilapia during the 120-day feeding trial, only male tilapia were selected and randomly allocated into three groups, with three pools per group and a density of ten individuals per pool. The fish from the three groups were randomly cultured in nine cement pools (2 m × 2 m × 1 m) in the Pearl River Fisheries Research Institute. There was no significant difference in the initial individual weight ($500 \pm 0.36$ g) between the three groups ($P > 0.05$). The fish were fed one of the three diets (EPD, PD, common diet) at 9:00 am and 5:00 pm each day for 120 days, and the feed amount for each day was $2.5 \pm 0.5\%$ of fish weight. The water temperature was kept at $25 \pm 2\,°C$, the pH was $7.0 \pm 0.5$, and the dissolved oxygen was above 5.0 mg/L.

## Sampling procedures

On the 120th day, six tilapia randomly captured from each group were placed into three big plastic containers (1 m × 1 m × 0.5 m) and euthanized with pH-buffered tricaine methanesulfonate (250 mg/L), and the rest of the surviving fish were cultured for future experiments. Once fin and operculum movement ceased, the body weight and body length were measured to calculate growth-related parameters. The fish were then sampled

randomly from each group. 10 mL of blood for each fish was taken from tail vein; five mL of blood was stored in an anticoagulant tube treated with EDTAK2 for hematologic analyses. The remaining five mL was left standing for 4 h, and the serum was separated by centrifugation at 3,500 r/min for 10 min, followed by immediate storage at −80 °C for further biochemical analyses (including immune, oxidative and antioxidative parameters). Next, the weight of the viscera and hepatopancreas were measured. Hepatopancreas tissues were sampled for the measurement of oxidative and antioxidative parameters. Some muscle and hepatopancreas samples of about $3 \times 3 \times$ three mm$^3$ were placed in 10% formalin for H&E staining. Midgut samples of 0.5 cm were fixed in Bouin's reagent for H&E staining. The remaining gut sample were collected for determining the activity of intestinal enzymes. The growth parameters were calculated as follows.

Weight gain rate (WGR, %) = (final weight − initial weight)/initial weight $\times$ 100

Condition factor (CF, %) = body weight/length$^3$ $\times$ 100

Visceral somatic index (VSI, %) = visceral weight/body weight $\times$ 100

Hepatopancreas somatic index (HSI, %) = hepatopancreas weight/body weight $\times$ 100

Survival rate (SR, %) = the number of surving fish/total initial number of fish $\times$ 100

Feed conversion rate (FCR) = total food intake/(final weight − initial weight).

The experimental protocols used in this study were approved by the Laboratory Animal Ethics Committee of Pearl River Fisheries Research Institute, CAFS, China, under permit number LAEC-PRFRI-20201118.

## Measurement of textural quality parameters

For texture analysis, fillets were taken from the back muscle. The textural quality parameters were examined using a Universal TA Texture Analyzer (Tengba, China) (*Ma et al., 2020a*). The parameters included hardness (g/f), chewiness (g/f), gumminess (g/f), adhesiveness (gf/mm), cohesiveness and springiness. Collagen content was determined by the Ultra-Sensitive Fish ELISA Kit (Sino, China) (Kit No. YX-E21992F). For the determination of muscle fiber diameter and density were measured as previously described in *Ma et al. (2020a)* and *Ma et al. (2020b)*.

## Measurement of biochemical indexes and enzymatic activities

Blood samples were treated with EDTAK2 (Sanli, China) within 4 h after being collected and were used to measure red blood cell count (RBC) with the methods described in previous methodology (*Natt & Herrick, 1952*). Counting was performed under a light microscope (Olympus, Japan). In addition, the Ultra-Sensitive Fish ELISA Kits were used to measure the indexes including serum haemoglobin (HGB) (Kit No. YX-E21998F), glucose-6-phosphate dehydrogenase (G6PD) (Kit No. YX-E21988F) and immune parameters including total protein (TP) (Kit No. YX-E21984F), albumin (ALB) (Kit No. YX-E21985F) and globulin (GLB) (Kit No. YX-E21986F).

The serum and hepatopancreas oxidative parameters were measured by the Ultra-Sensitive Fish ELISA Kits (Sino, China), including reactive oxygen species (ROS) (Kit No.

**Table 2  Growth parameters and muscle collagen content.**

|  | WGR (%) | CF (%) | VSI (%) | HSI (%) | SR (%) | FCR | Collagen content (μg/mg) |
|---|---|---|---|---|---|---|---|
| Control | 215.05 ± 1.32[a] | 3.77 ± 0.25[a] | 6.93 ± 0.24[a] | 1.83 ± 0.16[a] | 100 | 1.27[a] | 44.70 ± 0.94[b] |
| EPD | 218.22 ± 1.35[a] | 3.82 ± 0.21[a] | 7.13 ± 0.12[a] | 1.86 ± 0.17[a] | 100 | 1.25[a] | 73.46 ± 0.83[a] |
| PD | 216.33 ± 1.14[a] | 3.88 ± 0.23[a] | 6.98 ± 0.20[a] | 1.80 ± 0.18[a] | 100 | 1.26[a] | 66.23 ± 0.82[a] |

**Notes.**

Control, common diet group; EPD, expanded pellet diet group; PD, pellet diet group; WGR, weight gain rate; CF, condition factor; VSI, visceral somatic index; HSI, hepatopancreas somatic index; SR, survival rate; FCR, feed conversion rate.

Values of the same column with different letters were significantly different ($n = 6$, $P < 0.05$).

YX-E21809F), malondialdehyde (MDA) (Kit No. YX-E21989F), photohydrogen peroxide ($H_2O_2$) (Kit No. YX-E21807F), superoxide dismutase (SOD) (Kit No. YX-E21824F), glutathione (GSH) (Kit No. YX-E21817F) and nicotinamide adenine dinucleotide (NADPH) (Kit No. YX-E21969F). Absorbance was detected according to the previous methods (*Ma et al., 2020a*).

The intestinal enzymes activity (trypsin, amylase and lipase) were measured according to previous study (*Ma et al., 2020a*). Specifically, the intestine samples were homogenized in cooled saline, following centrifugation for 12 min (3000 rpm/min, 4 °C). Subsequently, the supernatant was collected to determine intestinal enzymes activity using the respective kits (Kit No. YX-E21975F, Kit No. YX-E21813F and Kit No. YX-E21965F) (Sino, China).

### H&E staining of muscle, hepatopancreas and intestine

The H&E staining of muscle, hepatopancreas and intestinal tissues was performed on according to the standard histology protocol (*Yu et al., 2017*). The morphological differences in intestine (three slides per fish) were observed using the following structures: lamina propria (LP), eosinophilic granulocytes (EG), mucosal folds (MF) and enterocytes nucleus (EN).

### Data analysis

Data were analyzed by one-way analysis of variance (ANOVA) using the SPSS 20.0 software (SPSS Inc., Chicago, IL, USA), followed by Bonferroni's test to determine the significant differences between groups ($P < 0.05$), and the p and F values were accurately calculated (95% confidence levels). The results are expressed as "mean ± SE". Statistical results are fully reported in File S1, including degrees of freedom, the exact F values, effect size and $P$-value.

## RESULTS

### Growth performance and muscle collagen content

The growth performance from three groups is shown in Table 2. The collagen content of the EPD and PD groups was much higher than that of the control group ($P < 0.05$). In addition, there were no significant differences in weight gain rates (WGR), condition factor (CF), visceral somatic index (VSI), hepatopancreas somatic index (HSI), survival rate (SR) or feed conversion rate (FCR) between the three groups ($P > 0.05$).
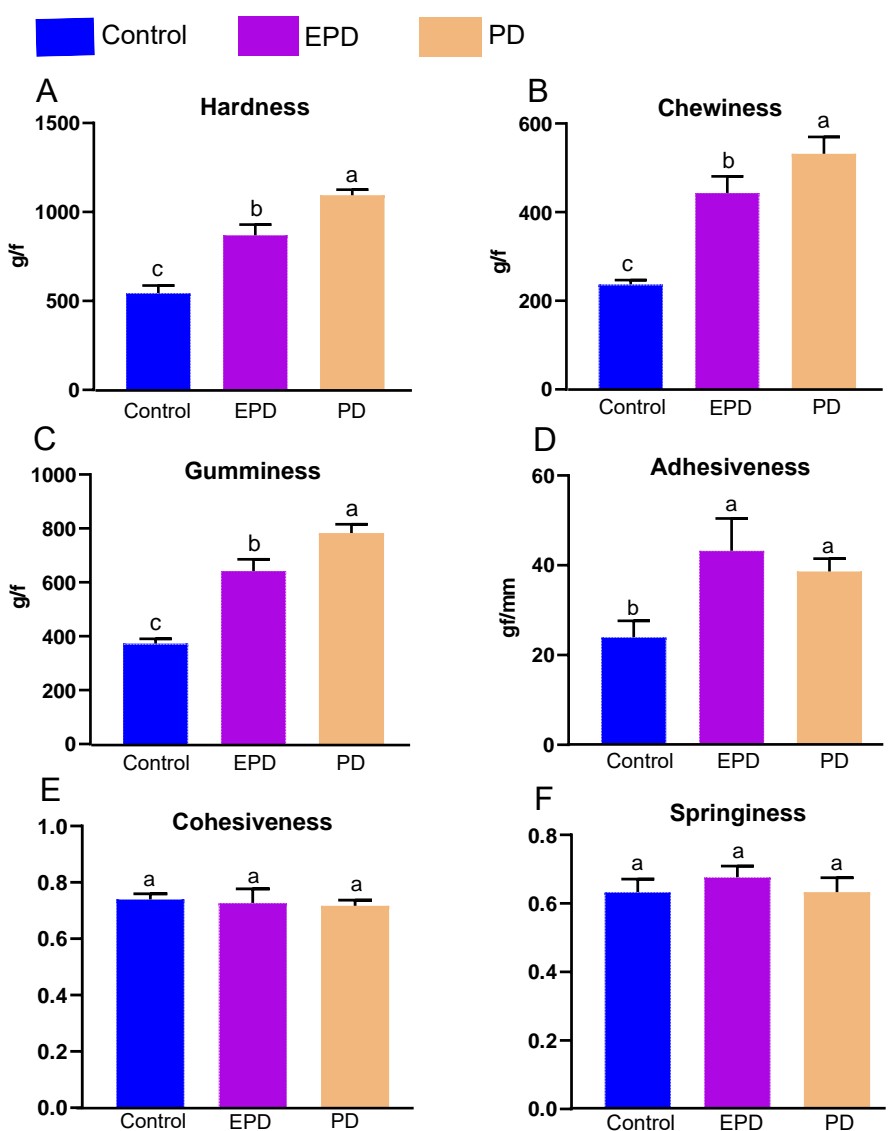

**Figure 2  Muscle textural quality.** Control, common diet group; EPD, expanded pellet diet group; PD, pellet diet group. Different letters were significantly different ($P < 0.05$).

## Textural parameters and microstructure observation of muscle

The textural parameters of the fillets, including hardness, chewiness, gumminess, adhesiveness, cohesiveness and springiness were measured for the three groups (Fig. 2). The hardness, chewiness, gumminess and adhesiveness of the EPD and PD groups were significantly higher than those of the control group ($P < 0.05$). Interestingly, hardness, chewiness and gumminess in the PD group were significantly higher than those of the EPD group ($P < 0.05$). Additionally, the cohesiveness and springiness in showed no significant difference between groups ($P > 0.05$).

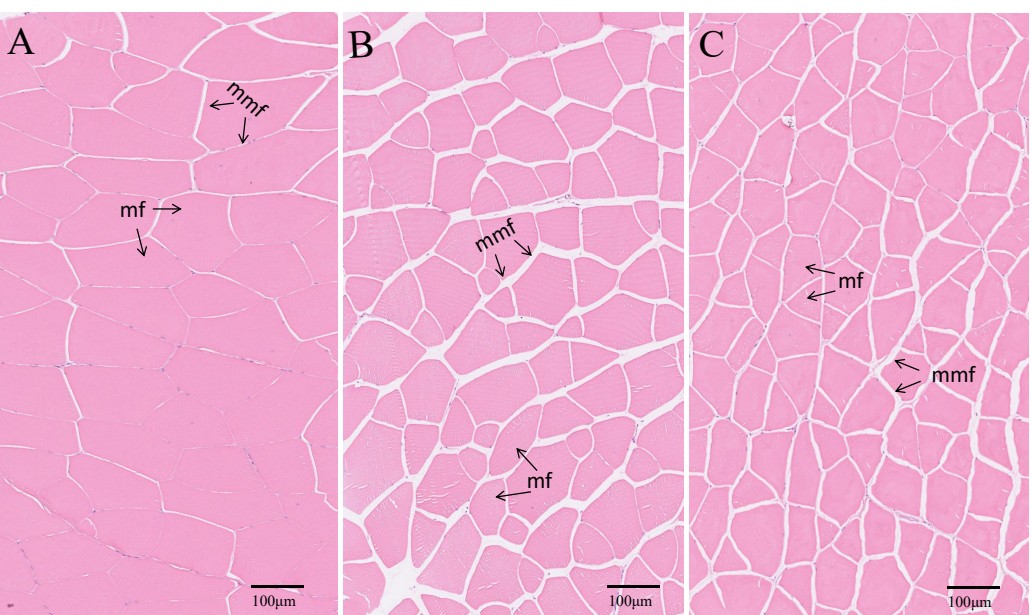

**Figure 3  Muscle transverse section microstructure.** (A) Common diet group; (B) expanded pellet diet group; (C) pellet diet group. H & E staining, bar = 100 μm. mf, muscle fiber; mmf, matrix between muscle fibers.

The transverse microstructure diagrams of muscle are shown in Fig. 3. Obviously, statistical analysis showed that the muscle fiber densities of the EPD ($108.0 \pm 1.0$ No./mm$^2$) and PD ($120.0 \pm 1.0$ No./mm$^2$) groups were significantly higher than that of the control ($74.0 \pm 1.0$ No./mm$^2$) ($P < 0.05$) (Fig. 3). In line with this, the EPD ($101.78 \pm 6.42$ μm) and PD groups ($91.97 \pm 7.51$ μm) exhibited lower muscle fiber diameters than did the control group ($175.68 \pm 13.21$ μm) ($P < 0.05$).

## Hematological indexes and immune parameters

The effects of different diets on hematological indexes and immune parameters were obtained (Fig. 4). There is no significant difference for red blood cell (RBC) counts ($P > 0.05$). For hemoglobin (HGB), the PD group was significantly lower than control group ($P < 0.05$), and EPD group was slightly higher than control group ($P > 0.05$) (Fig. 4B). The EPD and PD groups show significant lower activity of glucose-6-phosphate dehydrogenase (G6PD) ($P < 0.05$) (Fig. 4C). In addition, There is no significant difference for the contents of total protein (TP), albumin (ALB) and globulin (GLB) among three groups ($P > 0.05$) (Figs. 4D, 4E and 4F).

## Oxidative and antioxidative parameters of serum and hepatopancreas

In order to explore the effect of FB extracts on oxidative responses and antioxidative capabilities, the oxidative and antioxidative parameters were measured (Fig. 5). Specifically, the EPD and PD groups showed higher contents of ROS, $H_2O_2$ and MDA in serum than the control group ($P < 0.05$) (Figs. 5A, 5B and 5C), and the EPD and PD groups also showed higher contents of $H_2O_2$ and MDA than the control group in the hepatopancreas

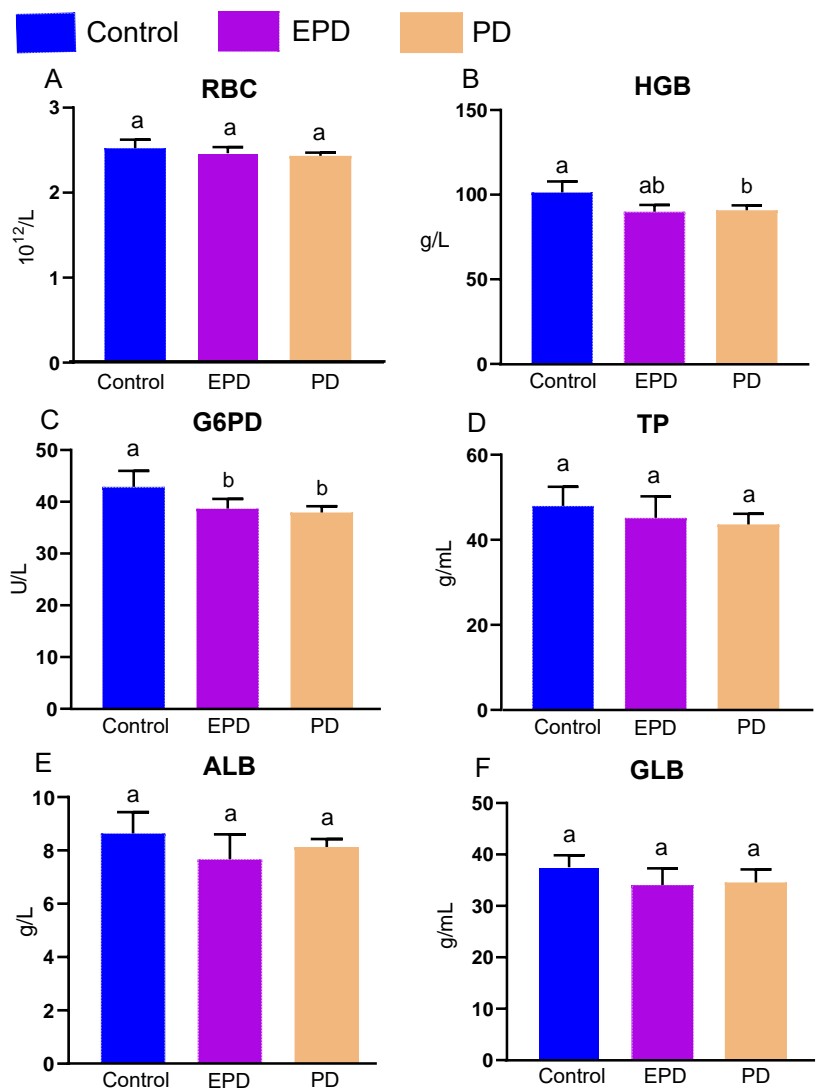

**Figure 4 Hematological indexes and immune parameters.** Control, common diet group; EPD, expanded pellet diet group; PD, pellet diet group. (A) Red blood cell counts (RBC); (B) hemoglobin (HGB); (C) glucose-6- phosphate dehydrogenase (G6PD); (D) total protein (TP); (E) albumin (ALB); (F) globulin (GLB). Different letters were significantly different ($P < 0.05$).

($P < 0.05$) (Figs. 5B and 5C). Interestingly, the EPD group showed a lower level of ROS than did the PD group ($P < 0.05$) (Fig. 5A). The antioxidative capabilities of serum and hepatopancreas were also significantly affected by diets (Figs. 5D, 5E and 5F). The activity of SOD in the serum of the EPD and PD groups was significantly lower than that of the control group ($P < 0.05$) (Fig. 5D). The control group showed higher levels of GSH and NADPH than did the other two groups ($P < 0.05$) (Figs. 5E and 5F), and the EPD group showed higher levels of NADPH than did the PD group ($P < 0.05$) (Fig. 5F). These results suggested that FB water extracts, to some extent, may cause mild oxidative damage to tilapia blood and hepatopancreas.

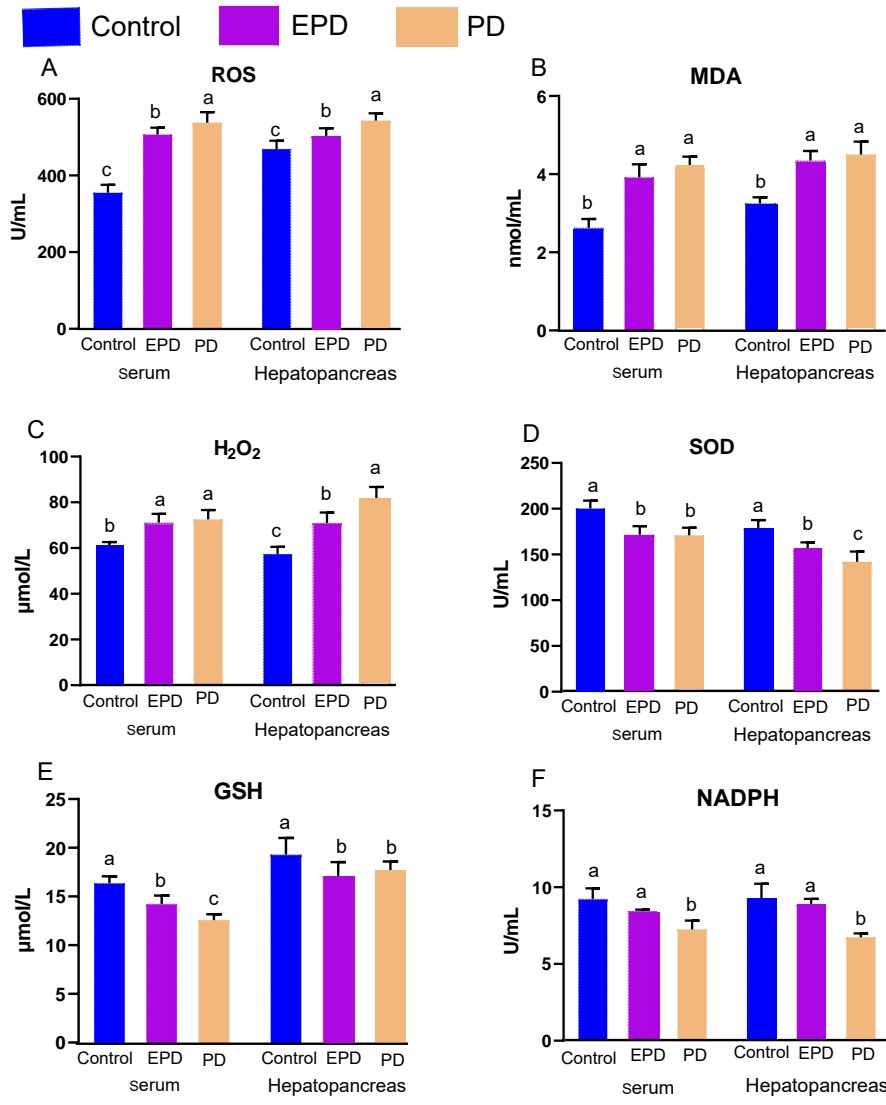

**Figure 5** **Oxidative and antioxidative parameters of serum and hepatopancreas.** Control, common diet group; EPD, expanded pellet diet group; PD, pellet diet group. (A) Reactive oxygen species (ROS); (B) malondialdehyde (MDA); (C) hydrogen peroxide (H2O2); (D) superoxide dismutase (SOD); (E) reduced glutathione (GSH); (F) reduced nicotinamide adenine dinucleotide phosphate(NADPH). Different letters were significantly different ($P < 0.05$).

## Histological structure of hepatopancreas

The hepatopancreas microstructures of the three groups were observed (Fig. 6). The number of nuclei (N) in the EPD group was the highest, followed by the control and PD groups (Fig. 6). More lipid vacuoles (LV) were found in the PD group (Fig. 6C), followed by the control and EPD groups (Fig. 6).

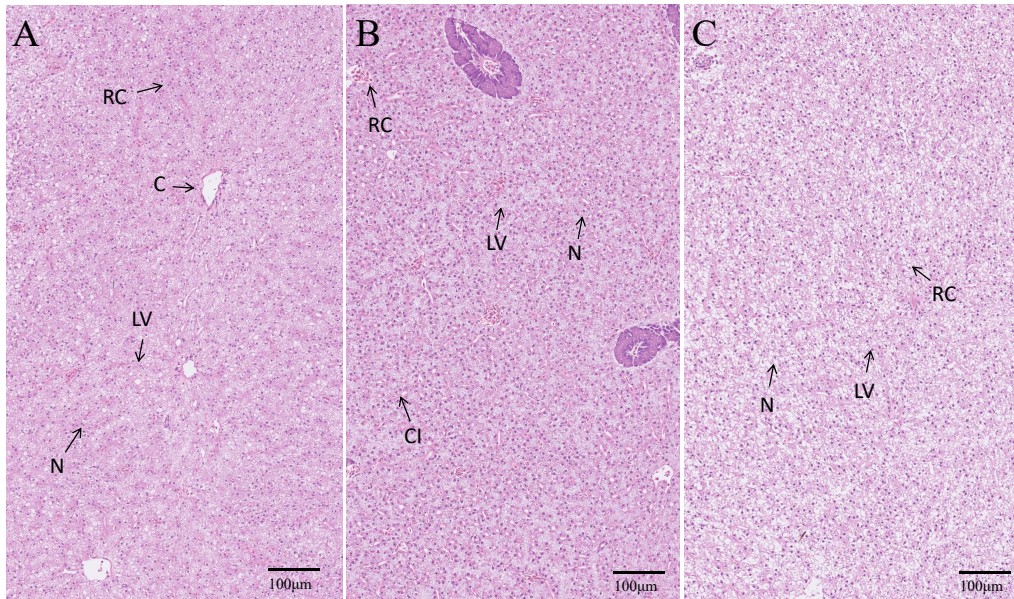

**Figure 6  Histological structure of hepatopancreas.** (A) Common diet group; (B) Expanded pellet diet group; (C) Pellet diet group. H & E staining, bar = 100 mm. Lipid vacuole (LV), central vein (C), nucleus (N), cellular infiltration (CI) and red cell (RC).

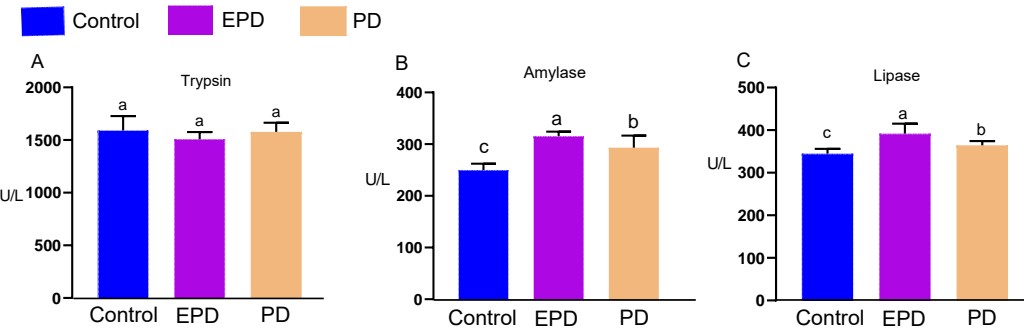

**Figure 7  The digestive enzymes of tilapia.** Control, common diet group; EPD, expanded pellet diet group; PD, pellet diet group. Different letters were significantly different ($P < 0.05$).

## Intestinal digestive enzymes and microstructure observation

The activities of three digestive enzymes (trypsin, amylase and lipase) were measured (Fig. 7). There was no significant difference in the activity of trypsin between the three groups (Fig. 7A) ($P > 0.05$). The changes in the amylase and lipase activity showed similar tendencies (Figs. 7B and 7C). The activity of amylase in the EPD and PD groups was significantly higher than that of control group (Fig. 7B) ($P < 0.05$). The activity of lipase in the EPD and PD groups also tended to be higher than that of control group (Fig. 7C). These results indicated that FB water extract exerted a positive influence on digestive enzymes.

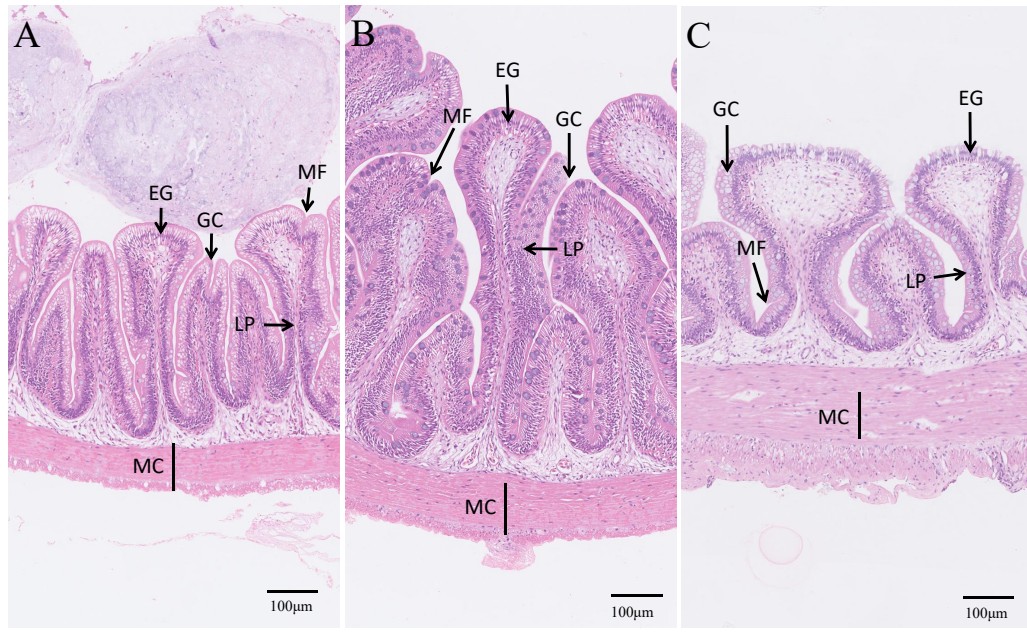

**Figure 8 Microstructure observation of intestinal tracts.** (A) Common diet group; (B) Expanded pellet diet group; (C) Pellet diet group. H & E staining, bar = 100 μm. LP, lamina propria; EG, eosinophilic granulocytes; MF, mucosal folds; MC, muscle thickness; GC, goblet cell.

Lastly, the intestinal microstructures of different groups were observed (Fig. 8). The intestinal villi of the control and EPD groups were smoother and flatter than that of the PD group, and the adjacent intestinal villi in the PD group were slightly damaged (Fig. 8), suggesting modest intestinal injury. Nevertheless, the muscle thickness of the intestine in the EPD and PD groups was increased compared to the control group (Fig. 8).

# DISCUSSION

## Texture quality improvement

Flesh quality issues have gained importance in the aquaculture industry, and texture is one of the most important quality indicators of fish products (*Valente et al., 2016*; *Ma et al., 2020a*). It has been reported that different FB extracts have been used to improve the muscle quality of tilapia, but these studies seldom demonstrated the effectiveness of the FB extract in textural quality improvement (*Chen et al., 2014*). Previous studies have proved that FB water extract could be used, as an aquafeed additive in the PD, to improve textural quality in grass carp (*Ma et al., 2020b*). However, no attempts have been made to apply FB water extract into the EPD to examine the stability of active compounds throughout the intense formulating process (120 °C). Thus, this study firstly formulated and applied an EPD containing FB water extract to tilapia culture and found that both the EPD and PD containing FB water extract improved the textural quality of tilapia. Yet, the PD group demonstrated a more significant effect related to textual quality than did the EPD group, suggesting that active compounds in the EPD were partially inactivated under

high temperature (120 °C). Therefore, we speculated that the active compounds are likely to be heat-sensitive. In a previous study, the vicine was recognized as an active compound (*Ma et al., 2020a*). However, this conclusion is challenged by the result of the present study since several studies have reported that vicine is heat-stable (*Cardador-Martínez et al., 2012*; *Patterson, Curran & Der, 2016*). Thus, exploring the active compounds will be the focus of the future work. Furthermore, it can be inferred that some strategies need be implemented to retain the biological efficacy of active compounds in FB water extract, such as supplying protective substances and optimizing manufacturing parameters during the formulation of the EPD. The next phase of study is to increase the dosage of FB water extract to enhance the efficiency of textural quality improvement in tilapia.

The improved texture quality mainly includes the hardness, chewiness and gumminess, *etc.* (*Ma et al., 2020a*; *Yu et al., 2019*). Increased muscle hardness is positively correlated with increased muscle fiber density and is negatively correlated with muscle fiber diameter (*Yu et al., 2017*). Increased muscle hardness is also well associated with elevated collagen content *via* the TGF-β/Smads signal pathway (*Yu et al., 2019*; *Xia et al., 2021*). Consistent with these previous findings, muscle hardness, muscle fiber density and collagen were increased in the Nile tilapia of the EPD and PD groups. These results further substantiates the concept that muscle hardness is tightly related to the muscle fiber density and collagen content in fish. Furthermore, these changes in muscle can probably be attributed to elevated ROS, because ROS not only enhances the proliferation of myofibers and increases the muscle fiber density of crisp grass carp by causing a spillover of cytochrome c, but also increases collagen contents by disrupting collagen turnover (*Yu et al., 2020*; *Chen et al., 2021*). The present study also demonstrated that muscle hardness improvement is positively correlated with elevated ROS in tilapia. However, the role of ROS in the muscle hardness of tilapia still requires further investigation. Additionally, both lipid and protein oxidations caused by ROS are one of the primary mechanisms for quality loss during the storage of fish (*Hematyar et al., 2019*). Thus, superfluous ROS in the muscle of the EPD and PD groups might exert undesirable effects on the quality of fish muscle product, which deserves further studies.

### Oxidative responses

A previous study has shown that feeding FB is an effective way to improve fillet textual quality. However, in grass carp fed FB, hemolysis is known to take place as in the case of "favism" (a potentially life-threatening acute hemolysis manifesting reduced RBC count caused by low-activity of G6PD by ingestion of FB), accompanied by reduced RBC count and decreased HGB content (*Yu et al., 2017*; *Gallo et al., 2018*). The present study found no obvious hemolysis in the EPD and PD groups, indicating that FB water extract mitigates the side effect induced by FB when used for improving the muscle textural quality in fish.

### Antioxidative ability and immune response

Hematological and hepatopancreas analysis is an important tool for evaluating the immune response and antioxidative ability of fish (*Fazio, 2018*). Hematological TP, ALB and GLB are the main proteins involved in fish immune response (*Misra et al., 2006*), and higher

contents of TP, ALB and GLB indicate stronger immunity (*Li et al., 2012*). In the present study, the contents of TP, ALB and GLB showed no significant difference, indicating that the FB water extract had no effect on the immunity of tilapia. In addition, the present results also showed that FB water extract weakened antioxidative ability (increased levels of ROS and decreased levels of SOD, GSH), suggesting that FB water extract may cause oxidative damage to some extent while improving the textual quality of Nile tilapia. Notably, the hepatopancreas histology of the EPD group demonstrated milder oxidative damage than that of the PD group, as also illustrated by decreased levels of ROS and $H_2O_2$, increased levels of NADPH. Thus, it is speculated that the intense formulation process (120 °C) of EPD have possibly facilitated the effective removal of the oxidative substances.

### Intestinal health

In fish, intestinal digestive enzymes (trypsin, amylase and lipase) play a critical role in the hydrolysis of protein, carbohydrate and lipid to form small absorbable units as energy or material for growth (*Chikwati et al., 2013*), and intestinal structural integrity is also closely related with fish health (*Li et al., 2018*; *Zhang et al., 2020*). There is evidence that feeding FB to improve textual quality causes decreased activity of digestive enzymes and intestinal injuries, reducing growth rate in grass carp (*Gan et al., 2017*; *Li et al., 2018*). Intestinal inflammation and damage were found in salmon (*Salmo salar*) fed FB proteins (*De Santis et al., 2015*; *De Santis et al., 2016*). In the present study, fish fed with the EPD demonstrated better intestinal integrity and higher activities of amylase and lipase compared to the other two groups, suggesting that the EPD is a more advisable option than PD in a commercial setting regarding both the health and quality of cultured tilapia. Overall, EPD should be feasible substitute for FB in tilapia culture.

### Potential influence of of oxidation on fish muscle

The hepatopancreas and intestine are the two primary digestive organs mainly responsible for the synthesis and secretion of digestive enzymes and the absorption of nutrients (*Hardy, 2003*). It has been reported that intestinal oxidative damage caused by feeding FB could be detrimental to the activity of intestinal enzymes, lowering the absorption of nutrients and the growth rate of grass carp (*Ma et al., 2020a*). In the present study, the PD group showed more serious oxidative damage in hepatopancreas and intestine, which might have affected the absorption of nutrients and then caused some difference in muscle composition from the EPD group.

## CONCLUSION

Overall, FB water extract improved the muscle textural quality(hardness and chewiness) and collagen content in Nile tilapia. Therefore, this study further demonstrated that the faba bean water extract could be used as a functional aquafeed additive to improve textural quality in freshwater fish. Although the textural quality (hardness and chewiness) in the PD group was better than that of the EPD group, the EPD containing FB water extract not only improved muscle textural quality in tilapia but also improved its health status by reducing the oxidative damage to the hepatopancreas and intestine compared to the PD

group. Together with the popularity of expanded pellet diet in commercial setting, EPD containing faba bean water extract could be ideal substitute for faba bean in tilapia culture, but the effectiveness of EPD could be enhanced by increasing the dosage of faba bean water extract or adding protective substances during the formulation of the diets. We will further explore the regulatory mechanism of faba bean active components (vicine, *etc.*) on improving muscle textural quality of tilapia.

### Funding
This work was funded by the National Key R&D Program of China (2019YFD0900303). The funders had no role in study design, data collection and analysis, decision to publish, or preparation of the manuscript.

### Grant Disclosures
The following grant information was disclosed by the authors:
National Key R&D Program of China: 2019YFD0900303.

### Competing Interests
The authors declare there are no competing interests

### Author Contributions
- Yichao Li conceived and designed the experiments, performed the experiments, analyzed the data, prepared figures and/or tables, authored or reviewed drafts of the paper, and approved the final draft.
- Junming Zhang and Bing Fu performed the experiments, analyzed the data, prepared figures and/or tables, authored or reviewed drafts of the paper, and approved the final draft.
- Jun Xie and Ermeng Yu conceived and designed the experiments, authored or reviewed drafts of the paper, and approved the final draft.
- Guangjun Wang performed the experiments, prepared figures and/or tables, resources, and approved the final draft.
- Jingjing Tian performed the experiments, prepared figures and/or tables, data curation, and approved the final draft.
- Yun Xia analyzed the data, authored or reviewed drafts of the paper, experimental instruction, and approved the final draft.

### Animal Ethics
The following information was supplied relating to ethical approvals (i.e., approving body and any reference numbers):

The Laboratory Animal Ethics Committee of Pearl River Fisheries Research Institute, CAFS, China.

## Data Availability

The raw data are available in the Supplementary File.

## Supplemental Information

Supplemental information for this article can be found online at http://dx.doi.org/10.7717/peerj.13048#supplemental-information.

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
