# Peer review of "Textural quality, growth parameters and oxidative responses in Nile tilapia (Oreochromis niloticus) fed faba bean water extract diet"

_PeerJ, doi:10.7717/peerj.13048_

## Round 0.1 · original submission · Major Revisions

Your manuscript has to be thoroughly reviewed. It is necessary to improve the English language to avoid repeating some unclear words or sentences.

The materials and methods section should be improved by adding all the necessary information to make the assays reproducible.

·

Basic reporting

The authors have conducted a novel and important study worthy of publication; however, a major revision will be recommended with a reconstruction of major parts of the manuscript.
Conceptual issues
The study was focused on formulating the feed mixture with an enhanced antioxidative potentials, but in the entirety of the manuscript, antioxidation was seldom discussed. The introduction has no description of the undesirable effects of oxidation to fish food quality. It might have been expected that the authors will refer to fish oils (such as omega-3-fatty acids) and give few sentences about their unsaturation can cause rancidity.
Furthermore, the studies conducted focused on intestinal organs. The authors did not mention how any oxidative damage to these organs (such as pancreas) could affect the final product, since most of these organs are removed as offal during the evisceration and degutting process. If there is any relationship between these organs and the final fish muscle (majorly eaten by the consumer) that can affect the quality of the latter, this should be well stated in the introduction and discussion aspects.
In addition, the difference between expanded pellet diet (EPD) and pellet diet (PD) were not clearly declared almost in the entire manuscript. Only lines 137 and 138 mention about using an expander. A paragraph in the introduction can be allotted to this where the difference between the two products are clearly stated, and why the pellets are normally expanded.

General correction
The use of the possessive pronoun “our” in line 105 and other parts of this manuscript should be removed. For example, instead of “our previous study…”, it can be “From previous studies…” Line 311 “Our next step…” to “the next phase of study is…”
The technical expression of making conclusions from other studies should be revised:
Line 89: “High stocking density exerted negative impact…” to “High stocking density was reported to exert negative impact…”
Line 91: ” …production did not bring high profit…” to :”…production was reported to not yielding high profit…”

Use of standard deviations instead of the ~ symbol
Lines 145 and 151: 2~3 to something such as 2.5±0.5
Line 151: 23~27 to something such as 25±2
Line 152: 6.5~7.5 to something such as 7.0±0.5

Other grammatical suggestions are shown below:
Line 55: “formulating…” to “formulation”
Line 98: “And its fillets…” to “Its fillets…” since this is a new sentence
Line 155: “On 120d…” to “On the 120th day…”
Discussion structure
The discussion should be decompartmentalized into simpler units of the results obtained with each unit being thoroughly discussed with recent and relevant literature.

Experimental design

Materials and methods reproducibility
The methods used for ROS, H202, and MDA results mentioned in line 260 were not declared in the materials and methods section
Authors did not mention key details to signify the originality and reproducibility of this study. Some questions left unanswered are listed below:
Line 127: “…adjusted pH to 9.0” with what acid (for an initially higher pH) or with what base (for an initially lower pH)?
Line 127: “…ultrasonic processing…” what kind of processing: probe or bath sonication? what are the sonication parameters like amplitude, frequency, and time?
Line 127: “…centrifugation…” what is the revolution speed? And for how long?
Line 128: “…concentrating the supernatant…” Through what separation process?
Line 133: “…dried at 50°C…” for what time duration?
Line 138: “…dried at 120°C…” for what time duration?

Validity of the findings

Study conclusion and recommendation
The authors mentioned in lines 366-370 that “we conclude that EPD… could be an advanced substitute…” thus offering a recommendation for it relative to the PD preparation. However, for textural parameters mentioned in lines 239-240, 301-302 that affect consumers choice and are indirectly the principal objective of the study, the PD preparation was reported to be “significantly higher than EPD”. EPD was marginally higher only for internal organs that might not be correlated with the final product. Besides, EPD requires further processing at a higher temperature of 120°C which could damage the product’s nutrients, cause rancidity, and certainly incur higher processing costs to the food vendor. The conclusion should be thoroughly revised for this.

Reviewer 2 ·

Basic reporting

The present study formulated the expanded pellet diet (EPD) and pellet diet (PD) containing faba bean (Vicia faba, FB) water extract (FBW), as a potential aquafeed additive to increase flesh texture, conducted a feeding trail, to evaluate the effect of FBW to improve the textural quality of tilapia. This is an interesting study, the findings obtained would be a good reference for the research and industrial practice of assessing product quality and acceptability. On the other hand, there are some aspects needed to be clarified.

Experimental design

1. Control diet was called commercial feed (L135), what does it mean? It was bought from certain company or produced based on the requirement of the research team? Is it a kind of EPD? Please make it clear.
2. It is suggested to add the composition data of FBW in Table 1.
3. The reason why 8% was added in the diet should be mentioned somewhere.

Validity of the findings

1. Since EPD and PD were applied in the present study, it is suggested to cite some references about the comparison of EPD and PD in the discussion part, to discuss the possible influence caused by FBW in the diet or the feed processing method.

Additional comments

1. The phrase “Growth parameters” appeared in the title, so in the abstract the growth performance results should be mentioned.
2. Line 145, why “male” tilapia was emphasized here? Or it should be mentioned earlier?
3. Line 230, it is better to say that treatment are higher or lower than control, instead of mentioning control first. Same in L237-239, L251, et al.

---

## Round 0.2 · Minor Revisions

Dear author,

Your paper still needs some minor revisions. Please see the indications provided by reviewer 1 (include a new figure and also the agricultural and/or industrial relevance and application of the study in the conclusion section).

Best regards,

·

Basic reporting

The manuscript has been revised and is having a better technical outlook. However, a few minor revisions can increase its presentation profile..

Experimental design

For a novel and technical work as this, a graphical summary figure is suggested as a figure 1. This can be cited in a suitable position in the materials and method section. Other figures can be renumbered accordingly.

Validity of the findings

The authors can state the agricultural and/or industrial relevance and application of the study in the conclusion aspect. One or few sentences regarding this, and also regarding any future prospects of the study can be added.

Reviewer 2 ·

Basic reporting

no comment

Experimental design

no comment

Validity of the findings

no comment

Additional comments

My previous comments have been well incorporated. There are two more suggestion listed below
1. Line 373 in revised manuscript (tracked changes), one “of” should be deleted.
2. The conclusion of abstract and the main body of the manuscript should be consistent. And it is suggested to put the sentence from Line 386-389 to the discussion part, instead of in the conclusion.

---

## Round 0.3 · accepted · Accept

I am pleased to confirm that your paper has been accepted for publication in PeerJ.

Thank you for submitting your work to this journal.